# Plasma Electrolytic Oxidation Coatings of a 6061 Al Alloy in an Electrolyte with the Addition of K_2_ZrF_6_

**DOI:** 10.3390/ma16114142

**Published:** 2023-06-01

**Authors:** Chaohui Tu, Xuanyu Chen, Cancan Liu, Deye Li

**Affiliations:** College of Materials Science and Engineering, Nanjing Tech University, Nanjing 211800, China; 202061203306@njtech.edu.cn (C.T.); 202161203207@njtech.edu.cn (X.C.); 202061203169@njtech.edu.cn (D.L.)

**Keywords:** thermal control property, Al alloys, plasma electrolytic oxidation, emissivity, solar absorbance

## Abstract

In this study, white thermal control coatings were produced on a 6061 Al alloy using plasma electrolytic oxidation (PEO). The coatings were mainly formed by incorporating K_2_ZrF_6_. The phase composition, microstructure, thickness, and roughness of the coatings were characterized using X-ray diffraction (XRD), scanning electron microscopy (SEM), a surface roughness tester, and an eddy current thickness meter, respectively. The solar absorbance and infrared emissivity of the PEO coatings were measured using a UV–Vis–NIR spectrophotometer and FTIR spectrometer, respectively. The addition of K_2_ZrF_6_ to the trisodium phosphate electrolyte was found to significantly enhance the thickness of the white PEO coating on the Al alloy, with the coating thickness increasing in proportion to the concentration of K_2_ZrF_6_. Meanwhile, the surface roughness was observed to stabilize at a certain level as the K_2_ZrF_6_ concentration increased. At the same time, the addition of K_2_ZrF_6_ altered the growth mechanism of the coating. In the absence of K_2_ZrF_6_ in the electrolyte, the PEO coating on the Al alloy surface predominantly developed outwards. However, with the introduction of K_2_ZrF_6_, the coating’s growth mode transitioned to a combination of outward and inward growth, with the proportion of inward growth progressively increasing in proportion to the concentration of K_2_ZrF_6_. The addition of K_2_ZrF_6_ substantially enhanced the adhesion of the coating to the substrate and endowed it with exceptional thermal shock resistance, as the inward growth of the coating was facilitated by the presence of K_2_ZrF_6_. In addition, the phase composition of the aluminum alloy PEO coating in the electrolyte containing K_2_ZrF_6_ was dominated by tetragonal zirconia (t-ZrO_2_) and monoclinic zirconia (m-ZrO_2_). With the increase in K_2_ZrF_6_ concentration, the L* value of the coating increased from 71.69 to 90.53. Moreover, the coating absorbance α decreased, while the emissivity ε increased. Notably, at a K_2_ZrF_6_ concentration of 15 g/L, the coating exhibited the lowest absorbance (0.16) and the highest emissivity (0.72), which are attributed to the enhanced roughness resulting from the substantial increase in coating thickness caused by the addition of K_2_ZrF_6_, as well as the presence of ZrO_2_ with higher emissivity within the coating.

## 1. Introduction

With technological developments in aerospace, the exploration of space is becoming increasingly important, and alloys such as aluminum (Al), magnesium (Mg), and titanium (Ti) are favored in the aerospace sector because of their relatively light mass and high specific strength and stiffness. Among them, Al alloys are frequently applied in the aerospace sector with the aim of managing heat exchange between the spacecraft and its surroundings while sustaining the spacecraft’s standard operational temperature. To achieve this, Al alloys are coated with thermal control coatings possessing specific optical and thermal properties, with white thermal control coatings being the most widely utilized type of coating due to their low absorption and high emissivity [1,2]. In the design of spacecraft thermal control coatings, solar absorbance α (wavelength range 0.2~3 μm) and infrared emissivity (wavelength range 5~50 μm) are the most important performance indicators. Generally, the emissivity of thermal control coatings surpasses 0.5, and based on their absorbance characteristics, thermal control coatings with absorbance greater than 0.5 are referred to as high-absorbance, high-emissivity thermal control coatings, commonly known as infrared-band thermal control coatings. Conversely, thermal control coatings exhibiting an absorption rate of less than 0.5 are categorized as low-absorption and high-emissivity thermal control coatings, often referred to as solar diffuse-reflecting thermal control layers [3,4].

The preparation methods of thermal control coatings are thermal spray [5,6], cold spray [7,8], sol–gel [9,10], anodic oxidation [11,12], electroless plating [13,14], magnetron sputtering [15,16], etc. Among them, thermal spray and cold spray methods exhibit inadequate thermal performance, while magnetron sputtering entails higher costs. Sol–gel, anodic oxidation, and electroless plating methods, although viable, have the potential to contribute to environmental pollution. Hence, the search for an alternative technology is imperative in order to overcome these limitations and challenges.

Plasma electrolytic oxidation (PEO), also known as microarc oxidation (MAO), is an innovative electrochemical surface treatment technology derived from anodic oxidation [17,18,19]. Compared with anodic oxidation, PEO exhibits favorable characteristics such as environmental friendliness, reduced processing time, and enhanced efficiency. It is frequently employed to generate thick oxide coatings on various metals, including Al, Ti, Mg, and Zr. Some researchers have explored the utilization of PEO in molten salt environments [20,21,22]. In recent years, the research on thermal control performance using PEO has gradually increased. Early on, the European Space Agency studied the thermal control coatings of AA2219 and AA7075 alloys [23,24]. These coatings were prepared using specialized alkaline solutions, the specific formulations of which were not publicly disclosed due to certain confidential reasons. In Wu’s report, the black PEO coatings produced via the in situ growth of Al alloys in a NaAlO_2_ electrolyte could reach solar absorbance values of >0.90 and IR emissivity values of >0.77 [25]. Wang et al. found that the emissivity of Al alloy base coatings could reach approximately 0.85 within the wavelength range of 8–20 μm at a temperature of 500 °C [26]. In their investigation, Kim et al. observed that the inclusion of yttrium-stabilized zirconium dioxide resulted in a 0.05 increase in emissivity [27,28]. Moreover, Yao et al. achieved an emissivity value of 0.99 by utilizing K_2_ZrF_6_ on titanium alloys [29,30,31], signifying the effective enhancement of coating emissivity through the addition of Zr, and K_2_ZrF_6_ was found to contribute to emissivity augmentation.

Al alloys are commonly used to prepare spacecraft shell materials. However, due to their inadequate thermal control performance, it becomes necessary to apply thermal control coatings on the surface of Al alloys to ensure proper functionality in space [32,33]. Most of the previous studies of PEO coatings on Al alloys are focused on high-absorption and high-emissivity thermal control coatings, and there are few studies on coatings with low absorption and high emissivity [34,35]. In contrast to Mg and Ti alloys, PEO-formed coatings on Al alloys typically exhibit a gray color, indicating their lack of low-absorption characteristics. Consequently, the introduction of additives becomes imperative to enhance the coating properties and achieve low-absorption capabilities.

In this study, we sought to investigate the effect of K_2_ZrF_6_ on the thermal control properties of PEO coatings. The primary focus was to develop thermal control coatings with low absorption and high emissivity. Additionally, we explored the impact of K_2_ZrF_6_ concentration on the structural aspects and growth mechanism of PEO coatings.

## 2. Materials and Methods

### 2.1. Coating Preparation

A 6061 Al alloy with dimensions of *Φ*35 × 4 mm was selected as the base material (nominal composition in wt.% included Mg 1.1, Fe 0.7, Si 0.6, Zn 0.25, Cu 0.2, and balance Al). The surface of each sample was polished with SiC papers up to 300, 600, and 1000 grit. Subsequently, the specimens were thoroughly cleaned through ultrasonic treatment in ethanol.

The base electrolyte was obtained using 18 g/L Na_3_PO_4_ and 3 g/L Na_2_SiO_3_. Then, 0 g/L, 5 g/L, 10 g/L, and 15 g/L K_2_ZrF_6_ were added. The microarc oxidation plant used in this study was a composite plant (PEO, JEOL, JSM-7900F, Tokyo, Japan) consisting of a pulse power supply, an electrolyte bath, and a cooling system. The PEO system utilized in this study was a composite setup (PEO, Nanjing Haolang Environmental Protection Technology, JCL-AMOZ10, Nanchang, China) equipped with a pulsed power supply, an electrolyte bath, and a cooling system. The PEO system was equipped with three separate power supplies, each capable of delivering a maximum output current of 30 A, and these power supplies could be used independently. The model is available in either constant-current or constant-voltage output. The temperature of the electrolyte was kept below 30 °C using a cooling and stirring system. The PEO parameters during the PEO process were set as follows: current density of 6 A/dm^2^, duty cycle of 20%, frequency of 800 Hz, and treatment time of 30 min. Based on K_2_ZrF_6_ content in the electrolytes, the samples were successively denoted as P-Zr0, P-Zr5, P-Zr10, and P-Zr15.

### 2.2. Coating Characterization

The morphology and elemental composition of PEO coatings were observed using scanning electron microscopy (SEM, JEOL, JSM-7900F, Tokyo, Japan) equipped with an energy-dispersive X-ray spectrometer (EDS, JEOL, JSM-IT500A, Tokyo, Japan). The coating phases were identified through X-ray diffraction (XRD, D/Max-2400, Tokyo, Japan) with Cu K_α_ radiation. The scanning angle was set from 10 to 90°, with a step size of 0.02°, a scanning speed of 10°/min, and a sweep angle of 4°. The porosity of the coatings was calculated using ImageJ software. The coating thickness was measured using an eddy current thickness meter (FMP20, Fisher, Kehl, Germany). The coating roughness was measured using a surface roughness tester (Ra200, Jingmere Technology Co., Ltd., Beijing, China). The coating color was analyzed using a colorimeter (LS171, Linshang, China), and its parameters were determined using CIE L*a*b*. In this context, L represents lightness; a indicates dark green (negative value) and pink (positive value), while b refers to blue (negative value) and yellow (positive value).

### 2.3. Coatings’ Thermal Control

Thermal control properties consist of absorbance α and emissivity ε. The absorbance α of PEO coatings in the 0.2–2.5 μm wavelength region was characterized using UV–Vis–NIR spectrophotometry (Perkin Elmer Lambda 950, Freehold, NJ, USA). The theoretical value of the total solar absorbance was calculated using Formula (1) [36].
(1)$$\alpha =1-\rho =1-\frac{\int_{250\;\mathrm{nm}}^{2500\;\mathrm{nm}}\rho S_{\lambda }d\lambda }{\int_{250\;\mathrm{nm}}^{2500\;\mathrm{nm}}S_{\lambda }d\lambda }\;$$
where α is the total solar absorbance, ρ is the spectral reflectance at the wavelength of 0.25–2.5 μm, and $S_{\lambda }$ is the spectrum intensity of solar irradiance.

The coatings’ emissivity ε was determined using Fourier transform infrared spectrometry (JASCO FT/IR-6100). The detailed test methods can be found in a previous study [37]. The thermal stability of the coatings was measured using a thermal shock test. The PEO-coated samples were subjected to a muffle furnace at a temperature of 500 °C for 2 min, followed by rapid immersion in water below 30 °C. This cyclic process was repeated for 80 cycles, and the surface condition of the coatings was meticulously observed and recorded.

## 3. Results

### 3.1. Voltage–Time Response

Figure 1 shows the voltage–time curves obtained during the PEO process with varying concentrations of K_2_ZrF_6_ additive, ranging from 0 to 15 g/L. The graph reveals a distinct trend whereby the breakdown voltage of the PEO system decreased with increasing K_2_ZrF_6_ concentration. This decline in breakdown voltage corresponds to the initial phase of the anodic oxidation, characterized by the generation of numerous bubbles and the initiation of the formation of the oxide coating. Subsequently, the process entered a second stage wherein the system became more acidic due to the electrolyte composition. Consequently, the termination voltage at the end of the first stage experienced a decrease. Additionally, the presence of K_2_ZrF_6_ induced the hydrolysis of some H^+^. However, as the reaction progressed, the rate of coating thickening exceeded that of dissolution, resulting in the formation of a PEO coating with appreciable corrosion resistance. Consequently, the subsequent voltage gradually began to rise, albeit at a significantly slower rate than the initial stage, signifying that the second phase was characterized by PEO spark discharge. During the third stage of the PEO process, it was observed that the process voltage increased when 5 g/L K_2_ZrF_6_ was added, compared with the absence of K_2_ZrF_6_. However, as the concentration of K_2_ZrF_6_ increased further, the process voltage exhibited a decreasing trend. This behavior suggests that the addition of a certain amount of K_2_ZrF_6_ reduces the electrolyte resistance. However, as the concentration of K_2_ZrF_6_ increased, the electrolyte resistance initially decreased and then increased. Notably, when K_2_ZrF_6_ was added, the working voltage displayed an initial period of stability, followed by a subsequent increase. This implies that distinct electrochemical reactions take place during these two periods. Nevertheless, after 30 min of treatment, the voltages of the four electrolytes gradually converged, indicating a minimal influence on the final voltage after the addition of K_2_ZrF_6_.

### 3.2. Effect of K_2_ZrF_6_ on Surface Structure and Microstructure

The thickness and roughness of the PEO coatings prepared on the surfaces of the 6061 Al alloy are shown in Figure 2. It can be seen that with the increase in K_2_ZrF_6_ concentration, the coating thickness and roughness significantly increased compared with those before the addition of K_2_ZrF_6_, which indicates that the K_2_ZrF_6_ in the electrolyte was fully involved in the formation of PEO coating and that there was a high coating growth rate in this electrolyte. The marginal difference in roughness between P-Zr10 and P-Zr15 can be attributed to disparities in the spark discharge process. Following the introduction of K_2_ZrF_6_, the spark discharge size on the surface of the specimens increased in comparison to that before the addition of K_2_ZrF_6_, owing to the influence of the zirconium-containing gel produced as a result of K_2_ZrF_6_ addition. Consequently, this led to an increase in roughness. Such changes in coating thickness and roughness were conducive to the enhancement of thermal control properties.

Figure 3 shows the changes in the Al alloy surface coating when the concentration of K_2_ZrF_6_ ranged from 0 to 15 g/L. It was found that as the concentration of K_2_ZrF_6_ increased, the bump and porosity of the 6061 Al alloy coating increased, resulting in a significant difference in the surface morphology. This change became more pronounced when the K_2_ZrF_6_ concentration reached 10 g/L. The presence of pores is closely linked to the discharge pathway of PEO. Consequently, the inclusion of K_2_ZrF_6_ augments the spark size, leading to larger discharge pathways and a nonuniform distribution state. As a consequence, the coating surface became rougher. The minimal variation in roughness observed between P-Zr10 and P-Zr15 can be attributed to the substantial thickness, which reduces the impact of pores and protrusions on surface roughness.

To further analyze the surface properties of the PEO coatings, the porosity (percentage of pore area) of the PEO coatings was calculated from micro/macro images. Additionally, the color of the PEO coatings was also quantified using CIE L*a*b*, as shown in Table 1. As the concentration of K_2_ZrF_6_ increased, the porosity of the coatings exhibited a corresponding increase. This could be attributed to the formation of numerous large-sized pores and an elevation in coating irregularities. Notably, the porosity escalated from 0.53% in P-Zr0 to 17.76% in P-Zr15. Hence, it can be inferred that the concentration of K_2_ZrF_6_ had a substantial influence on the surface structure of the coating, ultimately leading to the observed roughness enhancement. Furthermore, prior to the introduction of K_2_ZrF_6_, the coating on the 6061 Al alloy exhibited a darker color, as shown in Figure 4. As alumina is generally white, this discrepancy in color may be attributed to the trace elements within the alloy. As can be seen in Table 1, after the addition of K_2_ZrF_6_, L* increased significantly, while a* and b* decreased significantly. This observed alteration can be attributed to the augmented presence of zirconium-containing compounds, resulting in a progressively lighter coating color, while the trace elements in the base material were less influential on the coating color.

Table 2 shows the analysis of the surface elemental content of the coatings prepared after controlling the concentration of K_2_ZrF_6_. A comparison of the surface elemental content between the coatings without the addition of K_2_ZrF_6_ revealed that the color was significantly influenced by substrate composition. The main elements of the coating after adding K_2_ZrF_6_ were O, Al, Zr, F, Na, and K. Among them, Al originated from the substrate, while the other elements were derived from the electrolyte. Notably, the content of F increased with the increase in K_2_ZrF_6_ concentration, and the content of Al decreased. Furthermore, it was found that the electrolyte played a major role in the growth of the plasma electrolytic oxide film, while the oxidation of the Al substrate was minimal. This demonstrates that employing a concentrated zirconate electrolyte significantly mitigated the oxidation of the Al substrate itself during the PEO process. Consequently, this mitigation prevented the substrate’s composition from affecting the color of the PEO coating. Consequently, this led to the production of a coating with higher L* and enhanced coating absorption. Moreover, in line with prior studies, the oxide of Zr exhibited a white appearance. Therefore, it can be deduced that the color of PEO coatings is closely associated with the quantity and state of the Zr element present within these coatings.

The XRD patterns of PEO coatings prepared under different conditions are shown in Figure 5. Without the addition of K_2_ZrF_6_ in the electrolyte, the composition of the coating phase was mainly Al_2_O_3_. However, when K_2_ZrF_6_ was added, the diffraction peaks of tetragonal zirconia (t-ZrO_2_) and monoclinic zirconia (m-ZrO_2_) appeared in the spectrum, along with weak peaks of Al_2_O_3_. This indicates that with the addition of K_2_ZrF_6_, the composition of the coating phase was mainly t-ZrO_2_ and m-ZrO_2_, with a small amount of Al_2_O_3_, and the intensity of the diffraction peaks of t-ZrO_2_ and m-ZrO_2_ increased with increasing concentration of K_2_ZrF_6_. Clearly, the presence of Zr, O, and F elements in the film layer can be attributed to the electrolyte. This observation further substantiates that the growth process of the PEO coating on the introduced Al alloy is primarily characterized by the deposition of compounds derived from the electrolyte, with the composition of the resultant coating being minimally influenced by the substrate itself. Additionally, as the coating thickness increased, a significant reduction in the intensity of the Al substrate peak became evident, likely attributable to the shielding effect exerted by the coating.

Figure 6 illustrates the cross-sectional elemental distribution within the white PEO coating of the Al alloy. The elemental distribution revealed the presence of Zr, F, and Al within the coatings. This observation further supports the notion that the growth mechanism of the Al alloy PEO coatings involves a combination of substrate oxidation and the deposition of compounds originating from the electrolyte. Additionally, it is evident that the regions of the distribution of Zr and F elements in the coatings exhibited significantly higher brightness than the regions of Al elemental distribution. This observation serves as an additional indication that the growth process of the PEO coatings primarily entails the deposition of compounds from the electrolyte.

### 3.3. Effect of K_2_ZrF_6_ Addition on the Growth Behavior of Coatings

Figure 7 presents a cross-sectional comparison between Al alloys with and without the addition of K_2_ZrF_6_. In the images, the yellow dashed line represents the original surface of the base material. The majority of the coatings prepared without the addition of the K_2_ZrF_6_ electrolyte are located above the dashed line, as shown in Figure 7a. This indicates that outward growth was dominant in these coatings. However, upon adding K_2_ZrF_6_, the coating growth mechanism was a combination of both inward and outward growth, as illustrated in Figure 7b–d. Furthermore, with an increase in K_2_ZrF_6_ concentration, the proportion of inward growth in the coating increased. This outcome signifies that K_2_ZrF_6_ had a stimulating effect on the inward growth of the coating, with the thickness of inward growth increasing in accordance with the concentration augmentation. Consequently, this contributed to the overall increase in coating thickness. Furthermore, inward growth implies improved bonding of the coating. Consequently, it can be inferred that the growth pattern of Al alloys in phosphate systems is primarily characterized by outward elongation. However, the addition of K_2_ZrF_6_ induces a combination of inward and outward growth patterns, thereby enhancing the bonding strength of the coating.

Along with the relevant data, the growth pattern of the coating after the addition of K_2_ZrF_6_ is shown in Figure 8. Initially, a minute layer of Al_2_O_3_ film formed on the coating surface. Simultaneously, the electrolysis of the K_2_ZrF_6_ solution generated suspended Zr(OH)_4_ particles [38,39,40,41]. Subsequently, the voltage curve with time reached a plateau, signifying that the consumption of the substrate primarily contributed to coating formation. The presence of ZrO_3_^−2^, resulting from the decomposition of K_2_ZrF_6_, impeded substrate consumption and participated in the reaction, leading to the formation of ZrO_2_. Consequently, the Al_2_O_3_ content in the coating began to diminish, while the ZrO_2_ content increased, causing an accelerated thickening of the coating. Following a certain period of reaction time, equilibrium was attained between substrate consumption and electrolyte interaction. During this stage, the coating thickening rate stabilized, leading to the flattening of the time–voltage curve. Ultimately, the formed coating in this study predominantly comprised ZrO_2_, resembling the white PEO coating with thermal control properties, as illustrated in Figure 8b.

### 3.4. Effect of K_2_ZrF_6_ on the Thermal Control Properties of Coatings

Figure 9 shows the total reflection spectra of the PEO coatings prepared under different conditions. It is evident that the coating incorporating K_2_ZrF_6_ exhibited significant enhancement across the entire wavelength range (0.2~2.5 μm) in comparison to the PEO coating without K_2_ZrF_6_. This enhancement corresponds to a reduction in the absorbance α, with the spectrum demonstrating that P-Zr15 achieved the highest reflectance. The absorbance α values were calculated using Formula (1) and are summarized in Table 3, along with the emissivity ε values. It is notable that as the concentration of K_2_ZrF_6_ increased, a consistent decrease in absorbance α and an increase in emissivity ε were observed. These outcomes indicate that the addition of K_2_ZrF_6_ effectively reduced the absorbance α and enhanced the emissivity of the coating.

The thermal stability of thermally controlled coatings is also a key factor when used in environments with medium-to-high temperatures, thus limiting their practical application. In this experiment, we investigated the thermal stability of thermally controlled coatings by conducting 80 thermal shock tests at 20–500 °C. Figure 10 shows that the macroscopic morphology of the coating became slightly smoother, and the color of the coating slightly became darker (L* decreased from 90 to 89) after the thermal shock test, which could be explained by the factor of the change in particles on the surface of the coating. After the thermal shock test, the coating surface did not change significantly, and no surface cracks or coating flaking occurred. These experimental findings show that the thermal control coating had good thermal stability [42,43,44].

Based on the results of the above analysis, it is evident that the thermal control properties of PEO coatings are influenced by various factors. In terms of absorbance, the PEO coating applied on 6061Al alloy exhibited high absorbance when prepared using an electrolyte devoid of K_2_ZrF_6_ addition [45]. However, upon the introduction of K_2_ZrF_6_ into the electrolyte, the absorbance of the PEO coating increased, indicating the influence of the substrate on the absorbance. On the other hand, the addition of color additives can also optimize the absorbance of the coating. In this experiment, the addition of K_2_ZrF_6_ to the electrolyte changed the phase composition of the coating from the original Al_2_O_3_ to a whiter ZrO_2_, indicating that not only did it improve the absorbance of the coating, but it also suppressed the influence of the substrate on the absorbance of the coating.

On the other hand, the influence on the emissivity of a coating is multifaceted and encompasses several factors. Firstly, the composition of the coating itself plays a vital role in determining its emissivity, as emissivity is an inherent property of a substance. To achieve a high-emissivity coating, the primary constituents of the coating must possess elevated emissivity properties. In the present experiment, the addition of K_2_ZrF_6_ facilitated the transformation of the low-emissivity Al_2_O_3_ phase into a relatively high-emissivity ZrO_2_ phase, thereby enhancing the overall emissivity of the coating. Secondly, when the coating undergoes multiple high-emissivity phases, the emissivity varies across different wavelength ranges due to the diverse magnitudes of emissivity. Consequently, reactions such as stretching and vibration may occur when compounds composed of two or more elements form the coating, thereby augmenting the coating’s emissivity. In this study, the coating phase had limited diversity, resulting in a comparatively lower emissivity. Future investigations should consider introducing other elements to enhance the emissivity of this particular coating. In the current study, the coating phase was too homogeneous, resulting in a relatively low emissivity. The surface structure of the coating also had an effect on the emissivity, with bumps and pores forming on the surface causing a change in roughness and thus affecting the emissivity. This phenomenon arises because coatings with higher roughness possess a larger specific surface area, facilitating improved external energy radiation and, thus, increasing the emissivity. This elucidates why P-Zr15 had a higher emissivity than P-Zr5 for the same phase composition, but the coating roughness used in this study was still too low and should be increased as a means of improving emissivity in subsequent studies. Lastly, it is worth noting that coating thickness exerted a relatively minor influence on emissivity. Extremely thin coatings tended to display lower emissivity, and emissivity values fluctuated as thickness increased until it reached a threshold of approximately 20 μm.

## 4. Conclusions

The addition of K_2_ZrF_6_ fostered an accelerated growth rate of the PEO coatings. The roughness tended to stabilize with increasing K_2_ZrF_6_ concentration up to a certain value. The best performance of the PEO coating was achieved at a K_2_ZrF_6_ concentration of 15 g/L, when the thickness was 113.13 μm, the roughness was 3.22 μm, and the porosity was 17.75%.In conventional electrolytes, the presence of trace elements within the substrate matrix influences both the color and absorption characteristics of the coating. This influence is contingent upon whether the trace elements within the matrix contain coloring ions. However, the introduction of K_2_ZrF_6_ serves to mitigate the impact of trace elements on the coating. Consequently, the L* value of the coating exhibited a notable increase, rising from 71.69 to 90.53.The introduction of K_2_ZrF_6_ resulted in a considerable reduction in the P element content within the coating, approaching near-zero levels. This reduction facilitated the ingress of Na, F, and K elements into the coating matrix. Consequently, the coating predominantly comprised tetragonal zirconia (t-ZrO_2_) and monoclinic zirconia (m-ZrO_2_) phases, with their relative proportions increasing proportionally with the concentration of K_2_ZrF_6_. Furthermore, amorphous compounds containing F were also present within the coating structure. This phenomenon arises due to the dominant influence of electrolyte deposition in the formation of this particular coating.The coating exhibited low absorbance (α ≤ 0.20) and high emissivity (ε ≥ 0.68). As the concentration of K_2_ZrF_6_ increased, absorbance α decreased, while emissivity ε increased, which may be due to the combined effect of the phase and structure of the coating. The coating of P-Zr15 had the highest absorbance α (0.16) and the highest emissivity ε (0.71). These PEO coatings have potential applications in solar and infrared heating by optimizing the electrolyte.The addition of K_2_ZrF_6_ changed the growth mechanism of the coating. In the absence of K_2_ZrF_6_ in the electrolyte, the PEO coating on the Al alloy surface predominantly exhibited outward growth. However, with the inclusion of K_2_ZrF_6_, the growth pattern of the coating transitioned into a combination of outward and inward growth, with the proportion of inward growth progressively increasing alongside the escalating of K_2_ZrF_6_ concentration. The incorporation of K_2_ZrF_6_ led to favorable conditions that promoted the inward growth of the coating, consequently enhancing the bonding strength between the coating and the substrate. As a result, the addition of K_2_ZrF_6_ led to notable thermal shock resistance in the coating, thus demonstrating improved durability under thermal stress conditions.

## Figures and Tables

**Figure 1 materials-16-04142-f001:**
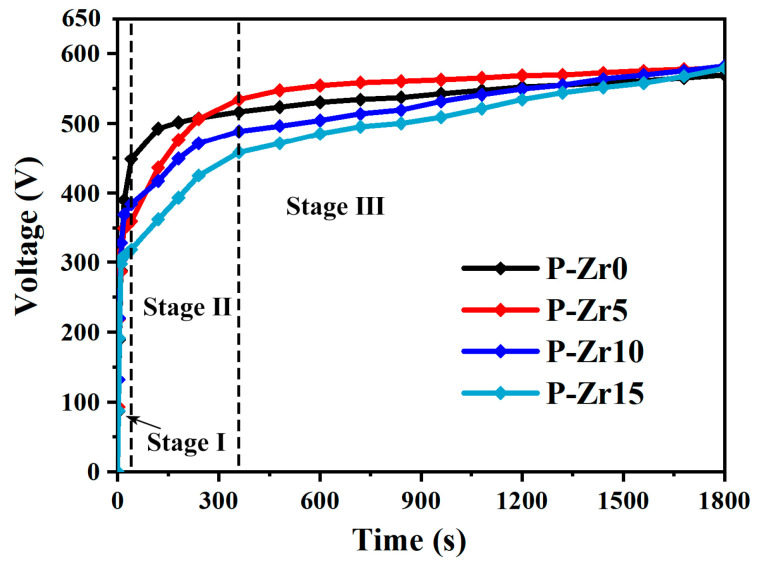
Time–voltage response curves for the PEO processes conducted in electrolytes with different concentrations of K_2_ZrF_6_.

**Figure 2 materials-16-04142-f002:**
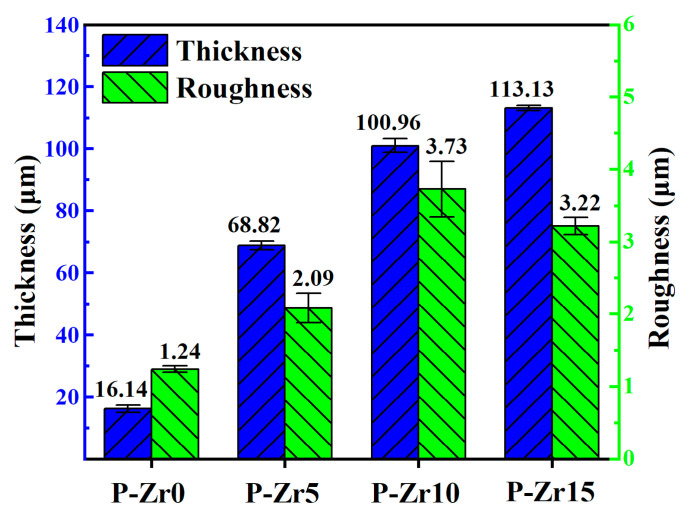
Thickness and roughness of PEO coatings with different concentrations of K_2_ZrF_6_.

**Figure 3 materials-16-04142-f003:**
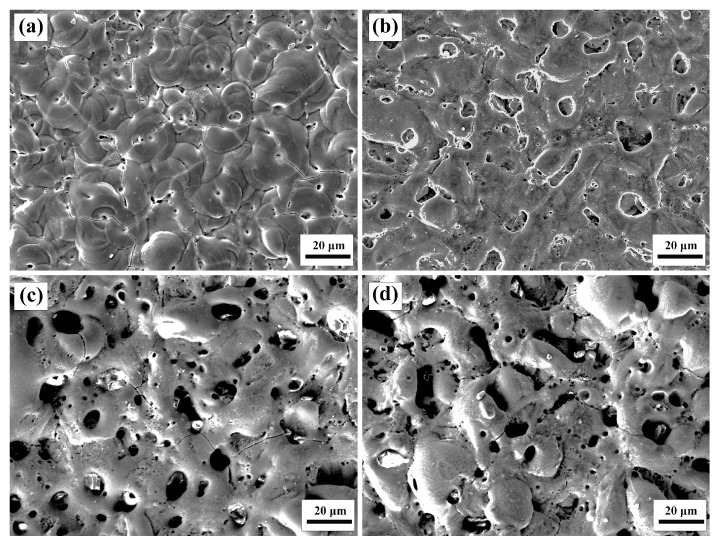
Surface SEM images of the PEO coatings obtained with different K_2_ZrF_6_ contents: (**a**) P-Zr0; (**b**) P-Zr5; (**c**) P-Zr10; (**d**) P-Zr15.

**Figure 4 materials-16-04142-f004:**
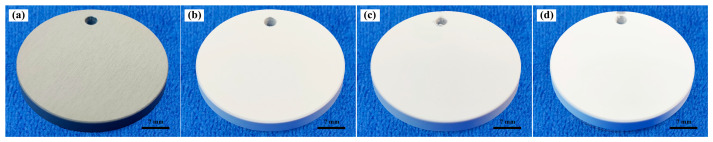
Comparative photographs of the surface color of PEO-coated samples: (**a**) P-Zr0; (**b**) P-Zr5; (**c**) P-Zr10; (**d**) P-Zr15.

**Figure 5 materials-16-04142-f005:**
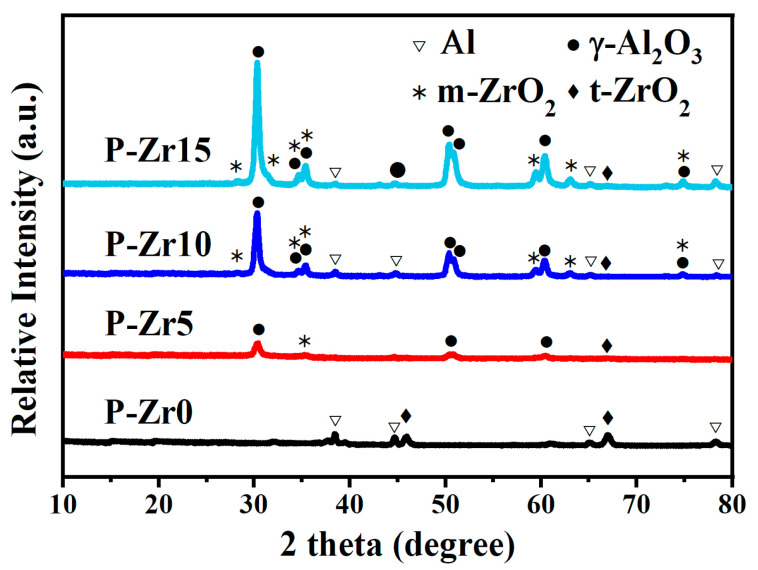
XRD patterns of the four PEO coatings formed in electrolytes containing varying quantities of K_2_ZrF_6_.

**Figure 6 materials-16-04142-f006:**
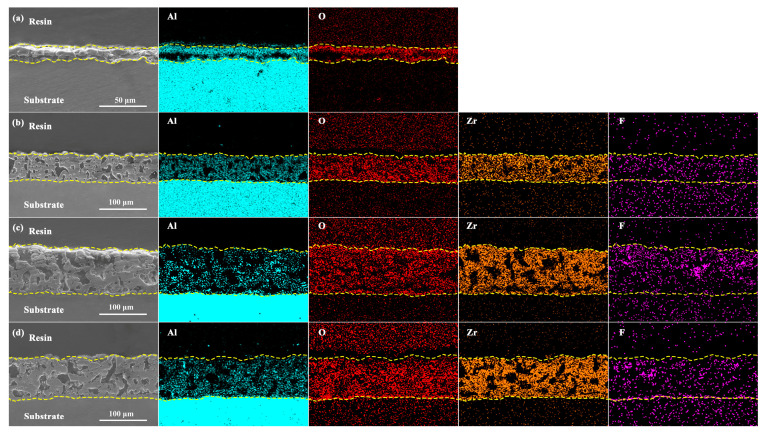
Cross-sectional morphology and cross-sectional elemental distribution: (**a**) P-Zr0; (**b**) P-Zr5; (**c**) P-Zr10; (**d**) P-Zr15.

**Figure 7 materials-16-04142-f007:**
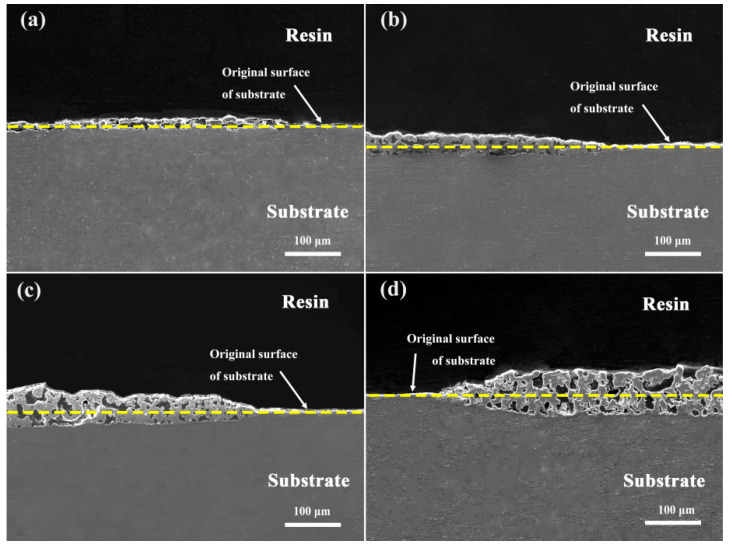
The growth characteristics of the coatings with and without the addition of K_2_ZrF_6_: (**a**) P-Zr0; (**b**) P-Zr5; (**c**) P-Zr10; (**d**) P-Zr15.

**Figure 8 materials-16-04142-f008:**
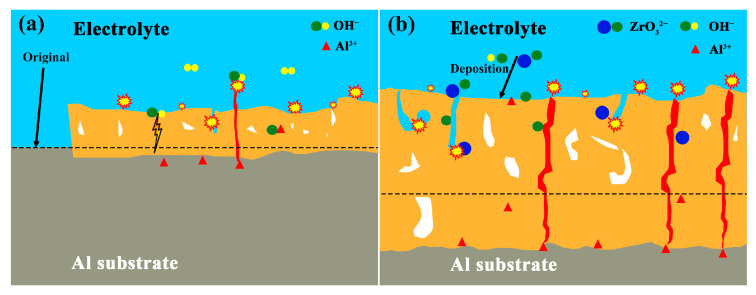
Schematic diagrams for the growth behaviors of the PEO coatings prepared in electrolytes without (**a**) and with (**b**) K_2_ZrF_6_.

**Figure 9 materials-16-04142-f009:**
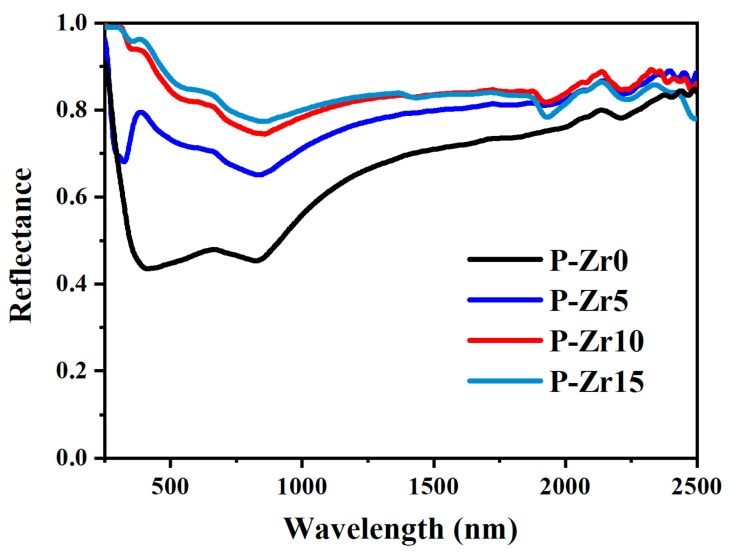
Solar reflection spectra of PEO coatings under different conditions.

**Figure 10 materials-16-04142-f010:**
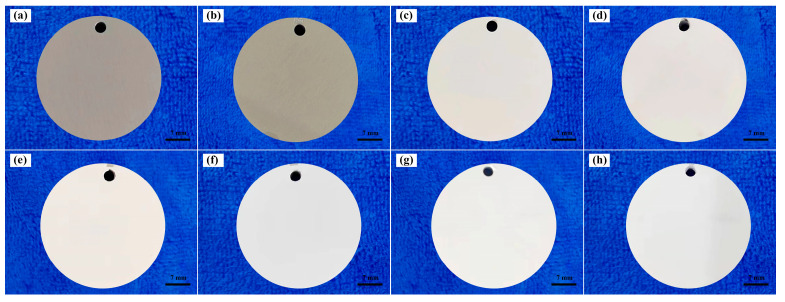
Comparison of the surface condition of the P-Zr15 coating before and after the thermal shock test; before the test: (**a**) P-Zr0; (**c**) P-Zr5; (**e**) P-Zr10; (**g**) P-Zr15. After 80 cycles: (**b**) P-Zr0; (**d**) P-Zr5; (**f**) P-Zr10; (**h**) P-Zr15.

**Table 1 materials-16-04142-t001:** Porosity and CIE L*a*b* results of PEO coatings under different conditions.

Specimens	Porosity (%)	L*	a*	b*
P-Zr0	0.53	71.69	1.44	5.57
P-Zr5	7.07	84.4	0.36	1.85
P-Zr10	11.95	89.33	0.61	1.06
P-Zr15	17.75	90.53	0.98	0.48

**Table 2 materials-16-04142-t002:** Elemental distribution of PEO-coated surface of 6061 Al alloy.

Specimens	Content of Elements (wt.%)							
	O	Al	P	Si	Na	Zr	K	F
P-Zr0	40.78	56.44	1.13	1.20	0.45	0	0	0
P-Zr5	21.64	19.66	0	0.31	3.58	39.87	4.51	10.43
P-Zr10	19.97	10.31	0.88	0.99	5.69	38.09	7.84	16.23
P-Zr15	20.62	10.35	0.01	1.04	5.19	42.11	7.52	13.16

**Table 3 materials-16-04142-t003:** Solar absorbance and infrared emissivity of PEO coatings.

Specimens	α	ε
P-Zr0	0.48	0.49
P-Zr5	0.27	0.68
P-Zr10	0.18	0.69
P-Zr15	0.16	0.71

## Data Availability

Not applicable.

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
