# Peer review of "Plasma Electrolytic Oxidation Coatings of a 6061 Al Alloy in an Electrolyte with the Addition of K2ZrF6"

_materials, 2023, doi:10.3390/ma16114142_

Round 1
Reviewer 1 Report
Dear Authors,
In this work, authors have synthesized an oxide coating on Al A6061 alloy using the plasma electrolytic oxidation (PEO) method in the electrolyte containing K2ZrF6. The authors have comprehensively investigated the morphology and composition of the coating, and thermal absorbance and emissivity.
The approach of this manuscript contains some novelty, experiments are carried out systematically, and the content of the manuscript is relevant to the scope of the journal. However, a major revision of the manuscript is needed for further consideration for publication. Specific comments are as follows:
1. I recommend mentioning follow works using an innovative approach of MAO in molten salt electrolytes: https://doi.org/10.1016/j.ceramint.2021.12.318; https://doi.org/10.1016/j.surfin.2022.102533;
2. Please specify the parameters of the XRD measurements used in the work, namely, geometry, X-ray source, electrical parameters, and scanning parameters. The name of the XRD device is not correct, please check.
3. It is clearly seen in Figure 1 that the breakdown voltage changes with the addition of K2ZrF6. Please add a short description of this point.
4. Please use the ‘PEO’ abbreviation for plasma electrolytic oxidation over the text.
5. I recommend deleting Section 4 Discussion because you discuss only the thermal absorbance and emissivity and not the overall obtained results. Just leave all the text without mentioning it as a new section.
6. I recommend adding results on the thickness, roughness, and porosity of the obtained coatings and mentioning the best performance in the Conclusions.
Reviewer 2 Report
The present work deals with the inclusion of K2ZrF6 as an additive in 6061 Al alloy by plasma electrolytic oxidation. The MS needs a revision of the language because some of the phrases are incomplete or even difficult to understand the main idea. Before its consideration for publication, the authors should take care of the next comments:
1) Line 161-164, please revise the information. The phrase seems to be incomplete.
2) Figure 4 is moved to the right side, so the scale is not shown properly. Also, avoid the use of a modified color background it may be misinterpreted as a full picture modification.
3) Line 193-194, revise the redaction of the sentence.
4) Figure 10 is moved to the right side, so the scale is not shown properly. Also, avoid the use of a modified color background it may be misinterpreted as a full picture modification.
5) Finally, the abstract must be revised. For the introduction section, the authors should show the novelty and objective of the work in a clearer way, and for the conclusion section, the information should be presented in a more critical way according to the main findings of the present work.
The MS needs a revision of the language because some of the phrases are incomplete or even difficult to understand the main idea
Reviewer 3 Report
There is no establishment of the phenomenological, nor an adequate discussion from the literature and the conceptual. The work is of interest, but it remains in the mere presentation of the basic results without delving beyond the data obtained. It is recommended to strengthen these aspects.
1-In the abstract, the following idea is repeated “reusability. The coating 14 becomes whiter with the addition of K2ZrF6”.
2-In the abstract to be more specific in the application when it says: “… indicating that the coating has promising applications for thermal 20 control coatings on Al alloys”.
3-Revise wording between lines 32-34.
4-Line 37, review typographical mistake.
5-Correct all spaces before references
6-The wording can be improved and spaces between paragraphs.
7-Why do the authors want to obtain low absorption and high emissivity thermal control coatings? It is not clear in the introduction.
8-Indicate the make, model and other data of the equipment used for the PEO process, as well as the parameters used for the characterization of the coating in each type of test performed.
9-Review the item 2.3. Coatings thermal control. The test conditions must be indicated.
10-The use of L*a*b in the text and then in table 1 is not very clear.
11-Revise the wording and clarify item 3.3
12-Is this subtitle correct? “3.4. Galling wear testing”
13-Line 265-266: “…and the absorbance α is 265 calculated according to formula”….state the formula.
14-Review wording lines 274 to 285.
15-How this was established: "After the thermal shock test, the coating 297 surface did not change significantly and no surface cracks or coating flaking 298 occurred. The experimental results show that the thermal control coating has good 299 thermal stability”
16-And the explanation of the other figures in “figure 10 Comparison of the surface condition of the P-Zr15 coating before and after the 302 thermal shock test (a) before the test (b) 80 cycles.”
17-You must cite everything that is required.
18-The discussion must be strengthened, well focused and well referenced. This is the main stuff to be organized.
Round 2
Reviewer 1 Report
Thank you for considering my comments.
Author Response
Thank you for your valuable and thoughtful comments.
Reviewer 2 Report
The authors have addressed the comments properly.
Please the grammar of some words along the MS, like " favourable". Which the correct word is "favorable".
Reviewer 3 Report
The article has improved according to the recommendations given.
Line 378, a typographical or punctuation error should be corrected.
